# An Experimental Study on the Influence of Drastically Varying Discharge Ratios on Bed Topography and Flow Structure at Urban Channel Confluences

**Zhiyuan Zhang** [1,2,3] **and Yuqing Lin** [3,4,*]

1   State Key Laboratory of Hydrology-Water Resources and Hydraulic Engineering, Hohai University, Nanjing 210098, China; zyzhanghhu@hhu.edu.cn
2   College of Water Conservancy and Hydropower Engineering, Hohai University, Nanjing 210098, China
3   Center for Eco-Environmental Research, Nanjing Hydraulic Research Institute, Nanjing 210029, China
4   State Key Laboratory of Hydrology-Water Resources and Hydraulic Engineering, Nanjing Hydraulic Research Institute, Nanjing 210029, China
*   Correspondence: yqlin@nhri.cn

**Abstract:** The confluences of rivers are important nodes for energy conversion and material transport in the river network. A slight morphological alteration of the confluences may trigger the "butterfly effect", which will bring about changes in the ecology and environment of the entire river network. During the transition period of the wet and dry seasons, the variation of discharge ratio will make the originally balanced river bed change again, which will bring a series of follow-up effects. This research mainly studied the features of water flow itself and results showed that the variation of discharge ratio caused secondary erosion of the balanced bed surface and transported the sediment downstream. Thus, the zone of maximum velocity was enlarged and the maximum flow velocity at the equal discharge was reduced, and more intense vortex and turbulence were generated. The lateral velocity, vertical velocity, and turbulent structure were mainly controlled by the quantity and ratio of the discharge, and the varying topography only played a minor role in local areas. Nowadays, some scholars have been studying the combination of flow field features and various environmental substances and biological habitats, and the basic work done in this article has laid the foundation for these studies.

**Keywords:** river confluence; discharge ratio; flow structure; secondary scouring

## 1. Introduction

It is well known that the change of topography at river confluences affects the flow structure, which often controls the material transport and energy exchange at the nodes of the river network. A remarkable process has been made by related studies over the past four decades. Best's conceptual model of confluence hydraulics has been widely recognized, and applied and the research directions and hotspots are usually focused on the following six zones: regions of flow stagnation, flow deflection, flow separation, maximum velocity, flow recovery and shear layers [1,2]. Low velocities and vortices [3–5] in the flow separation zone tend to concentrate pollutants in the river network in this zone [6]. The maximum velocity gradient and eddy structure [7] of the shear layer bring more intense material and energy exchange [8]. There are more abundant food resources on both sides of the shear layer, and it is the place where many large fish frequently appear in the natural river [9–11]. Therefore, a better understanding of the hydraulic characteristics in channel confluences is of guiding significance for the ecological management and environmental restoration of the entire river network.

Flow structure and bed topography are two major research hotspots in confluence hydraulics. In the study of water flow structure, it is difficult to explain the complex hydraulic characteristics of the intersection by the time-average flow rate alone due to

the collision of water and violent energy conversion caused by different flow directions, and more knowledge about turbulence is needed to reveal the mechanism of channel confluences [12–21]. Best found the distortion of the mixed layer at the intersection of channels at different depths. This distortion of the mixed layer will enhance vertical fluid upwelling and affect the mixing speed of the river [12]. Biswal et al. found the influence of discharge ratio on secondary flow stress and turbulence stress in the intersection zone [13]. Guillen Ludena et al. found that the discharge ratios and junction angles are the main controlling factors for the mountain river confluences [14]. Leclair et al. studied the variability of sedimentary structure and bed morphology at the junction of low-flow inconsistent rivers [15], proving that the deposition and erosion zones at the junction are related to the location of the shear layer, and illustrating the influence of flow structure on bed topography. Constantinescu et al. simulated the large-scale turbulent structure at an asymmetrical river confluence using an eddy-resolving numerical model [16]. In addition, the research on the bed topography in the confluence area is another hot topic in recent years [22–26]. Guillen Ludena et al. studied the influence of changes in hydrodynamic conditions on riverbed morphology brought about by local tributary widening with 70-degree confluence, and found that local widening of downstream reaches of tributaries would significantly enhance the non-uniformity of riverbed morphology [22]. Their research also found that the bed morphology of uniform sediments showed attenuation characteristics compared with that of non-uniform sediments [23]. Boyer et al. found that the inconsistency of bed height in the confluence area increased the turbulence intensity and enhanced the upwelling of the water in the confluence, indicating the influence of bed topography on the flow structure [24]. However, the flow conditions in these studies were usually constant or slightly variable, and insufficient attention was paid to the interaction between flow structure and bed topography.

Changes in the discharge ratio of natural rivers and artificial channels occur frequently, especially in plain areas and coastal cities with dense drainage systems [27,28]. The sudden increase of the tributary flood caused by local flood discharge or local rainstorm will make the confluence ratio vary sharply in a short time. In addition, 90-degree confluence is very common in nature [29,30] and is sensitive to topographical changes, so it can well reflect the changes in bed topography and flow structure caused by varying discharge ratios. Nevertheless, how to evaluate the comprehensive impact of this drastically varying discharge ratios on the riverbed topography and flow structure in the confluence area is still lacking in research.

In this study, we conducted experimental simulations through an indoor 90-degree flume, using a laser rangefinder to measure the temporal evolution of bed surface between two conditions: (1) the discharge ratio was changed drastically during a short period of time to simulate the sudden increase in tributaries when a flood strikes; (2) the discharge ratio was changed during a short period of time and then restored to simulate the conventional situation after the tributary flood fades. The 3D flow field and vortex structure were measured using an Acoustic Doppler Velocimeter (ADV) along 14 cross-sections under three different flow fields. This paper aimed to study the variation of bed topography dominated by discharge ratio and its reverse influence on the flow structure at a 90-degree channel confluence.

## 2. Experimental Equipment and Measurement

The experimental device is a 90-degree open channel confluence with automatic circulation system to simulate a common confluence situation in natural rivers. The water circulation system is mainly composed of the following parts: three water tanks; a PVC pipe with an inner diameter of 110 mm; two pipe pumps that control the mainstream and tributary, respectively; a transparent glass channel; a tail water valve; and a honeycomb rectifier. The general design and specific parameters of the flume channel are shown in Figure 1.

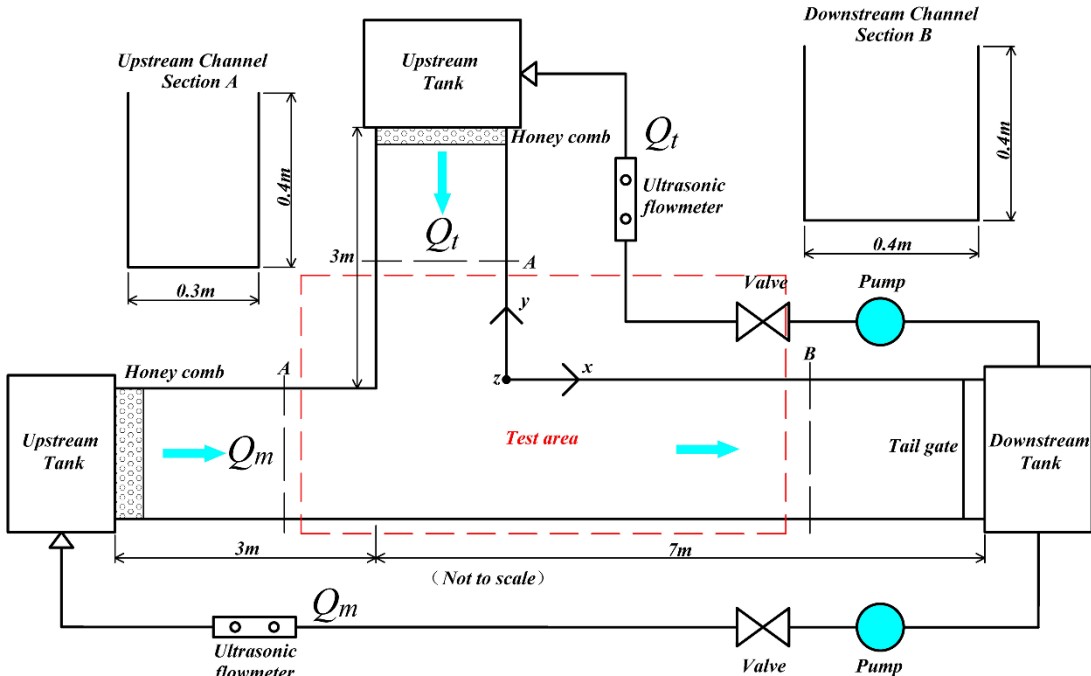

**Figure 1.** Schematic diagram of the experimental device: a 90-degree confluence channel flume.

The test was conducted using coarse sand with a median particle size of $D_{50}$ = 0.95 mm, and one 10 cm layer of sand was evenly spread on the bottom of the entire channel to simulate a natural riverbed. The water level was controlled at 26 cm by the tail water valve. The discharge ratio of natural rivers is in the range of 0.27–0.68 and it was approximately 0.4 and 0.6 in this study.

Two terrain cases and three flow field cases were considered.

Case 1: The mainstream discharge $Q_m$ and the tributary discharge $Q_t$ were 17.2 and 11.5 L/s, respectively. The discharge ratio [$R_q = Q_t/(Q_m + Q_t)$] was approximately 0.4, which was recorded as flow A, and the bed surface was basically balanced after 20–24 h of bed scouring. Then, $Q_m$ and $Q_t$ were changed to 11.5 and 17.2 L/s, respectively; the discharge ratio was approximately 0.6, which was recorded as flow B. 24 h scouring was performed under the condition of flow A and B in turn. The topography was fixed and measured, and it was recorded as topography B. The flow field under the condition of topography A and flow A was measured and recorded as flow field A.

Case 2: Flow field B was measured under the condition of topography A and flow B.

Case 3: Scouring was performed under the condition of flow A, B, and A in turn. The topography was fixed and measured, and it was recorded as topography B. Flow field C was measured under the condition of topography B and flow A.

The topography and flow field of the confluence zone were intensively measured, ADV (NorTek Vectrino+; maximum velocity: 4 m/s; sampled volume resolution: ±1 mm/s; sampling frequency of 100 Hz over a duration of 120 s) was used to measure the three-dimensional flow field, and a laser rangefinder (Extech DT500) was used to measure the topography. Weak signals (<30 signal-to-noise ratio) and poor correlation signals (<90% correlation) velocity data were removed. We processed the velocity data with a despiking method, and did not replace the removed data. The velocity data and topography data were recorded and then post-processed by Tecplot 360 software. After interpolation, sectional velocity maps and 3D topographic maps were drawn. The calculation formula of turbulent kinetic energy $k$ is as follows [29]:

$$k = \frac{1}{2}\left(u'^2 + v'^2 + w'^2\right)$$

where velocity fluctuations, $u'^2, v'^2, w'^2$, were calculated by the variance of velocity:

$$u'^2 = \frac{1}{n}\sum_{1}^{n}(u_i - \overline{u_i})^2$$

where $u_i$ is the component of velocity, $u$, $v$ and $w$ ($i$ = 1, 2, or 3), and their mean values are represented as $\overline{u_i}$; $n$ is the number of velocity data.

The shear layer was roughly visualized using colored dyes for the measured section to be as perpendicular as possible to the shear layer. A schematic diagram and the point coordinates of endpoints of measured sections are shown in Figure 2. Measured points were spaced every 2–3 cm horizontally and 1 cm vertically.

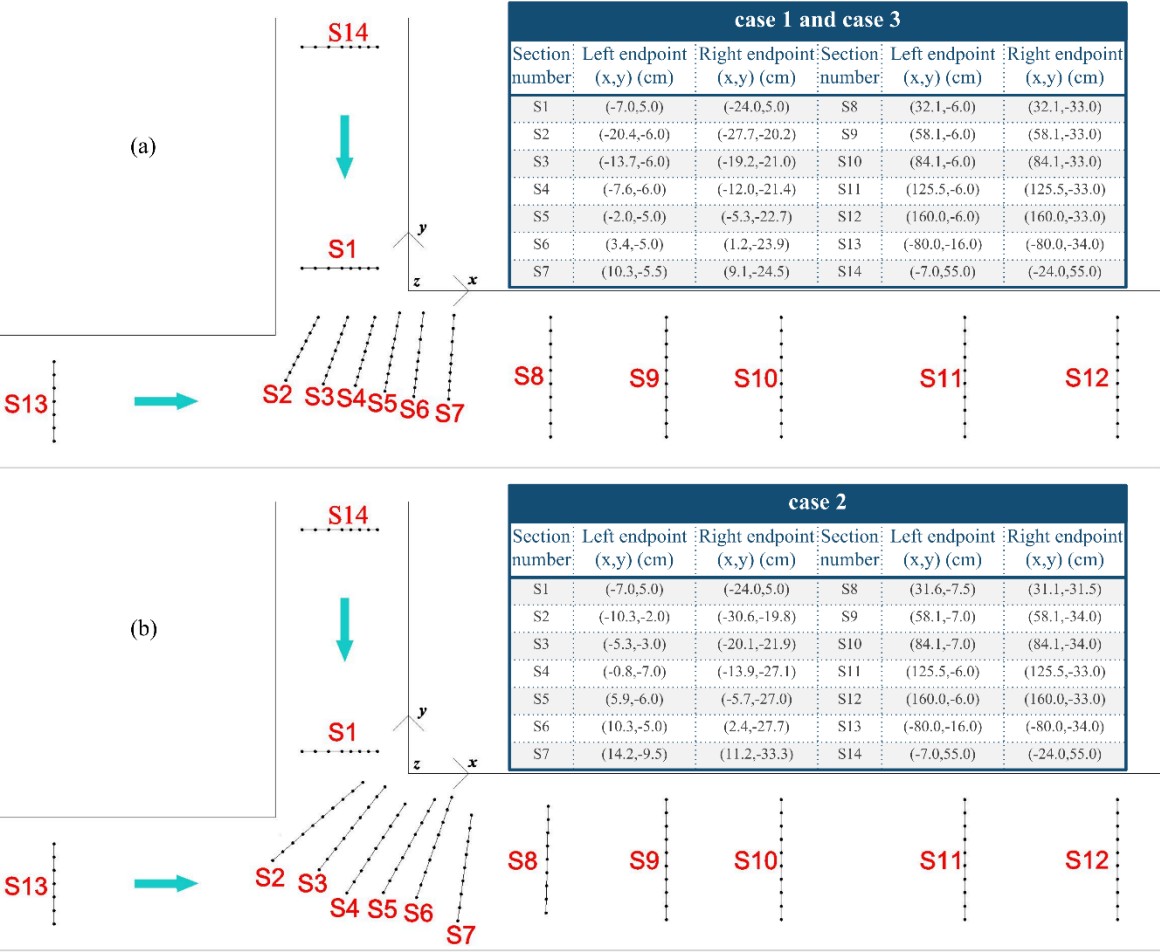

**Figure 2.** Schematic diagram of measurement cross sections. (**a**) case 1 and 3, $R_q$ = 0.4; (**b**) case 2, $R_q$ = 0.6.

## 3. Results

### 3.1. Bed Topography

Figure 3 shows the different bed topographies under two different cases, with the original bed surface as the reference plane.

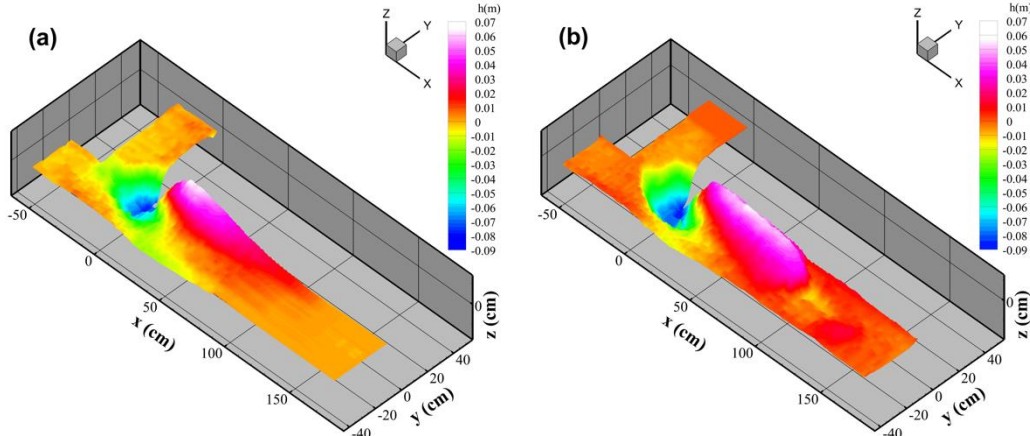

**Figure 3.** Three-dimensional bed topography of three cases. (**a**) case 1 and 3; (**b**) case 2.

In topography A, the maximum scour depth was about −7.5 cm, the highest point of the sand bar in the separation zone was around 6.5 cm, and the maximum drop was approximately 14 cm. The sand hole mainly surrounded the downstream intersection angle and the sand bar in the separation zone extended to the downstream intersection angle of 100 cm, with the maximum width of approximately 20 cm, accounting for half of the channel width.

In topography B, the maximum scour depth was approximately −8.5 cm, the highest point of the sand bar was around 5.5 cm, and the maximum drop was approximately 14 cm. The area of the sand hole was expanded. The maximum extension distance of the sand bar was around 100 cm, with a slight change, but the maximum width was expanded to approximately 30 cm. A local sand hole with a depth of 1–2 cm was generated at 110–130 cm downstream and a local sand ridge with a height of 1–2 cm was generated at 150 cm downstream, which was located on the right bank of the channel. Compared with topography A, a downward movement could be found on the whole, which indicating a secondary scour due to the effect of abrupt discharge on the topography. The sand in the original sand hole and sand bar was brought up and moved downward under the force of changing discharge, and a local uplift was formed after a part of the sand sank in the flow recovery zone. The variable stress caused by the change of discharge ratio remained unclear.

### 3.2. Three-Dimensional Velocity Field

The cross-sectional flow field distribution of the streamwise velocity U is shown in Figure 4. Negative U velocity was generated in the upper part of the separation zone near the downstream junction angle, which could be observed at approximately the position of cross sections 6 and 7 and is related to the horizontal vortex formed there. With the continuous expansion of the separation zone along the downstream, the flow area is also constantly shrank, resulting in the maximum flow velocity zone, which produces the maximum flow velocity near sections 8–10. In topography A, the maximum velocities of flow fields A and B were around 42 cm/s, while that of flow C was only 37 cm/s. This difference was due to the secondary scouring of the sand bar in topography b, which expanded the cross-current section occupied by topography and reduced the maximum flow velocity by approximately 12%. At sections 11 and 12, the two streams become basically comparable in the flow recovery zone. Sections 13 and 14 verified that the upstream flow was uniform under the guarantee of the rectifier. However, at section 1, the velocity distribution of the section exhibited slight decrease from left to right, and a negative velocity was observed near the bed surface. Under topography A, the shear layer in flow field B was distorted counterclockwise compared to A [29]. However, this phenomenon almost disappeared under the same flow and different topography cases.

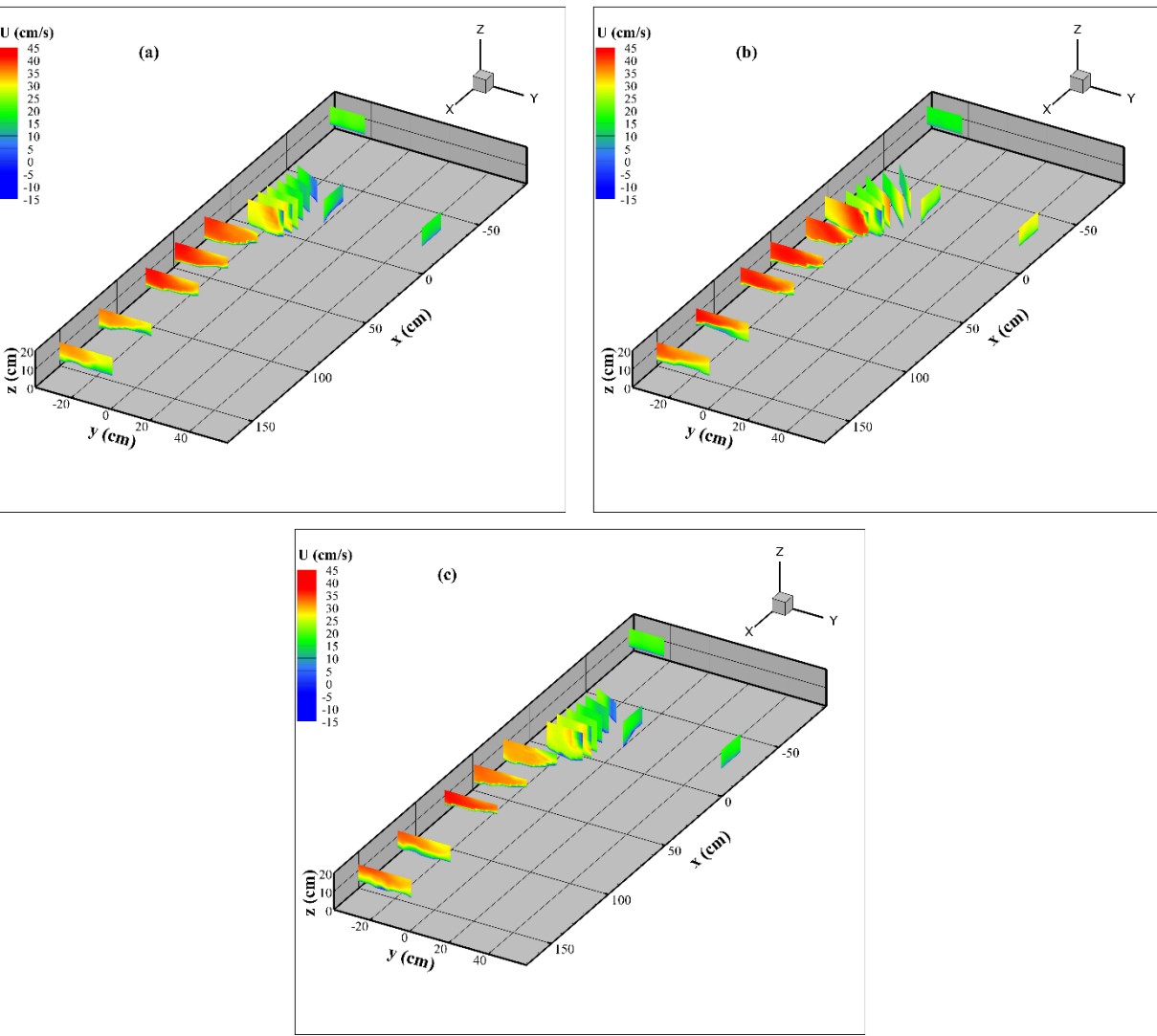

**Figure 4.** The spatial distribution of streamwise velocities U of three cases. (**a**) case 1; (**b**) case 2; (**c**) case 3.

The cross-sectional flow field distribution of the lateral velocity V is shown in Figure 5, which is the horizontal velocity component parallel to the cross section and it is positive when it points to the left of the main channel. Although the topographical conditions of flow fields A and C differed, they almost had the same flow field V distribution under the same discharge ratio. The maximum positive value and the minimum negative value were approximately 13 cm/s and −19 cm/s, respectively; however, those in flow field B obviously differed at around 19 and −17 cm/s, respectively. A negative value was also observed at the beginning of the separation zone, which is also related to the water surface vortex in this zone. A large negative value was found near the sand hole on bed surface, and the lateral flow was violent. Similar to that of U-field, the V-field distribution of section 1 exhibited abnormal inhomogeneity. In addition, it should be noted that at section 8, the lateral velocity above the sand bar in flow field C was roughly twice as large as in flow field A, where the highest topography was located.

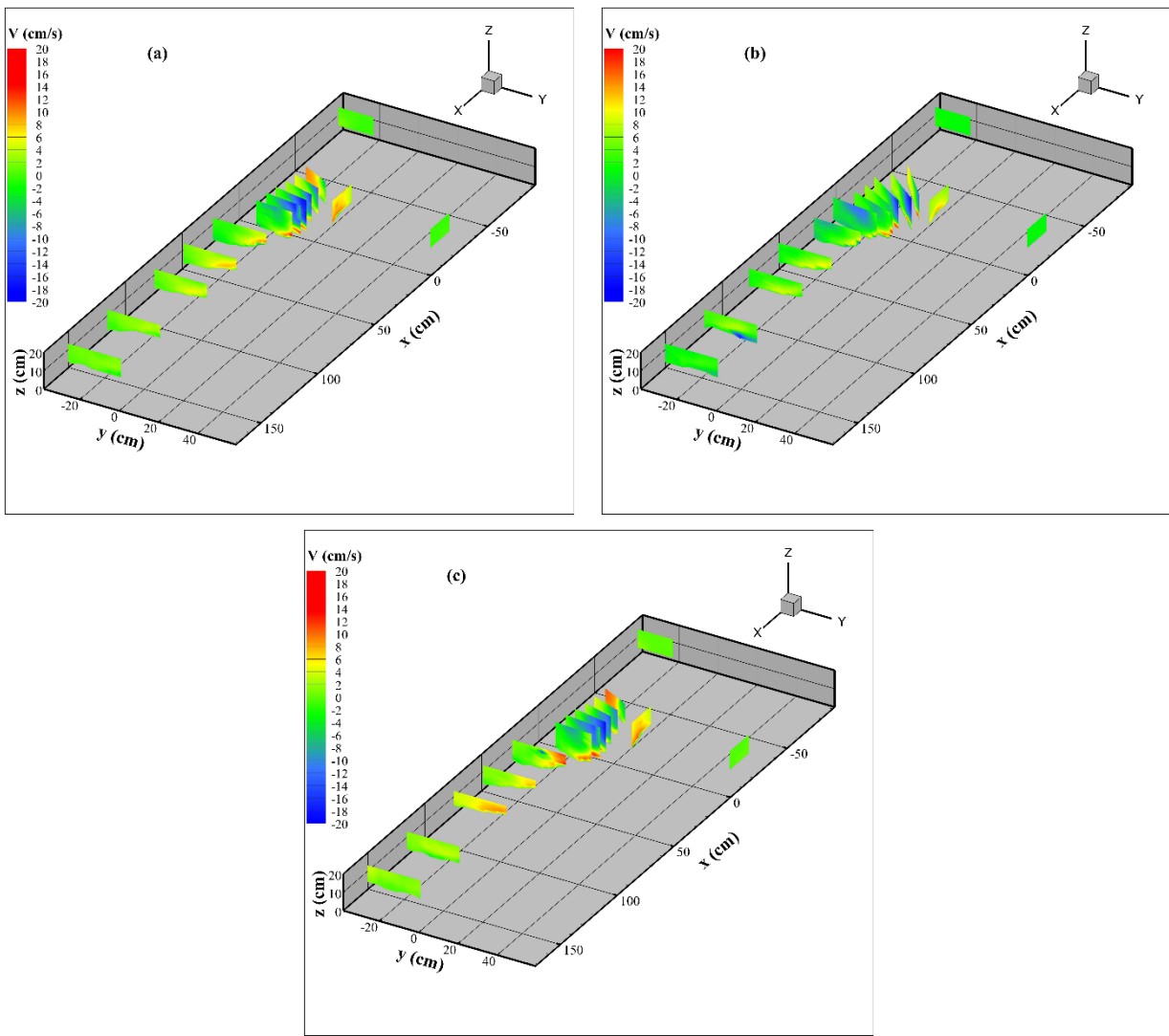

**Figure 5.** The spatial distribution of lateral velocities V of three cases. (**a**) case 1; (**b**) case 2; (**c**) case 3.

The cross-sectional flow field distribution of the vertical velocity W is shown in Figure 6. The minimum negative values appeared on cross sections 2–6 of the core area of the confluence and in the zone at a certain distance above the bed surface, which was consistent with the flow structure of the scour hole topography. The flow field distribution of A and C was basically the same; the minimum value was approximately −9 cm/s, but the maximum value of A (approximately 17 cm/s) was 2 cm/s larger than that of C (approximately 15 cm/s). The minimum value of flow field B reached −13 cm/s, nearly 50% higher than the other two, and the sand bar in the separation zone showed wider upward velocity distribution. Similarly, due to the change of topography, flow field C produced a more obvious vertical vortex above the sand bar at section 8 with a negative value, and the central area of sections 11 and 12 showed a larger negative value zone. This counterclockwise vertical vortex also existed in the separation zone near the downstream junction angle, and a spiral vortex was formed after mixing with the horizontal vortex, mainly distributed on both sides of the shear layer [29].

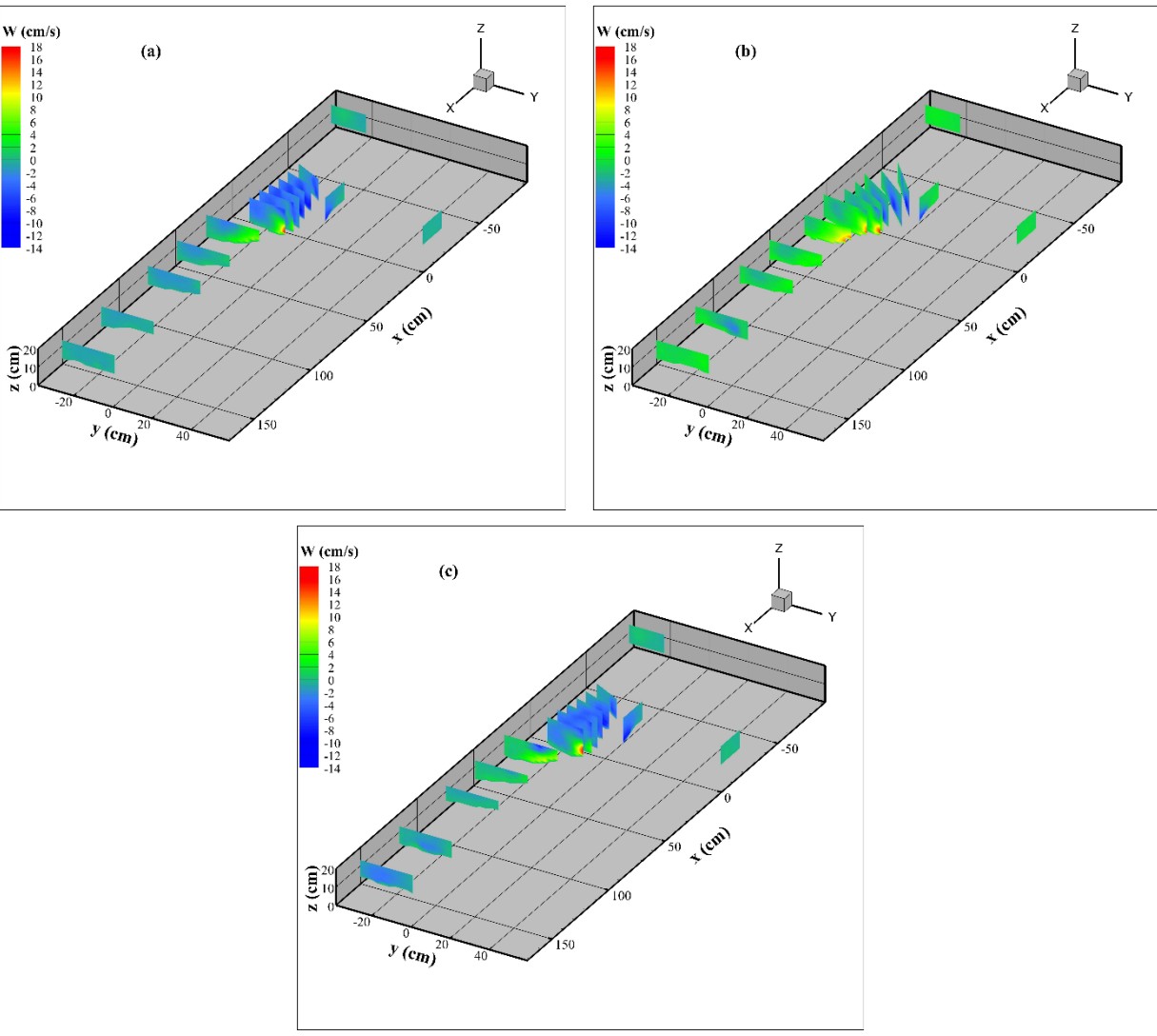

**Figure 6.** The spatial distribution of vertical velocities W of three cases. (**a**) case 1; (**b**) case 2; (**c**) case 3.

### 3.3. Turbulent Kinetic Energy

Figure 7 illustrates the cross-sectional flow field distribution of turbulent kinetic energy *k*, which can reflect the time average structure of turbulence. The overall k-field of flow field B was about 1.5 times of that of A, especially in the flow separation zone, where the turbulent flow was more severe. The results showed that the turbulence became more obvious under the condition of flow dominated by tributary. The k-field of flow field C was similar to that of A, but section 8 exhibited an abnormal value of *k* ($cm^2/s^2$), reaching approximately 240, which was twice the maximum value of A. This finding was also consistent with the facts observed in the V-field and W-field. The positions of sections 11 and 12, respectively, corresponded to the local sand hole and local sand ridge, respectively, and the near-bed surface *k* of field C in these two sections was approximately twice that of field A. These data all showed the influence of topography on the turbulent kinetic energy. In addition, the counterclockwise distortion of the shear layer was able to be evidently monitored in field B, but it was hardly reflected in C, which was consistent with the previous analysis, indicating that the influence of topographical conditions on the shear layer is secondary.

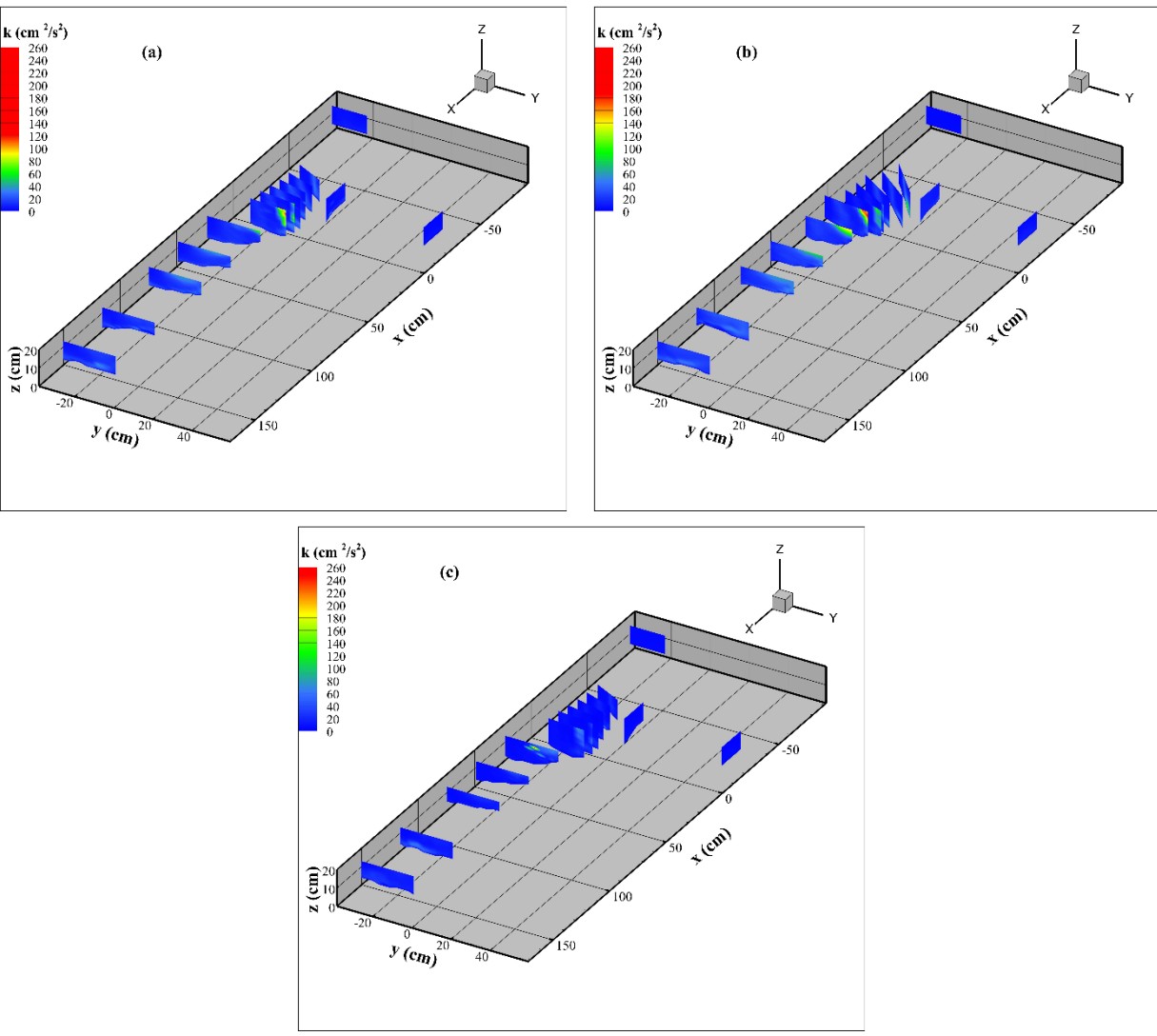

**Figure 7.** The spatial distribution of turbulent kinetic energy *k* of three cases. (**a**) case 1; (**b**) case 2; (**c**) case 3.

## 4. Discussions

In open channel confluences, the variation of flow conditions has a significant impact on the bed topography. Different scholars have conducted many related studies from different perspectives, which are mainly reflected in the flow depth [12], local width [22] and width-depth ratio [31] of flumes, junction angles [14], bed surface inhomogeneity [24], discharge ratios [14] and so on. Among them, the impact of discharge ratios on bed topography is mainly concerned with mountain rivers [14]. To our best knowledge, there is insufficient research on the varying discharge ratios of plain river networks due to flooding or the flow adjustment of water conservancy facilities, and the reverse influence of riverbed topography on flow structure is rarely revealed. Our research showed that varying discharge ratios had a linkage effect and a comprehensive impact on bed topography and flow structure. The results showed that the topography of the confluence area had a secondary scouring when the confluence ratio was reversed for a short time in the 90-degree confluence channel dominated by the mainstream flow. The bedload of the sand hole and the sand bar in the separation zone were brought up and moved downward under the force generated by the changing flow, so that the depth of the hole increased and the height of the bar decreased, and the local sand hole and sand ridge were formed in the flow recovery zone. We speculate that this variable stress caused by the reversion of the discharge ratio mainly occurred mainly in the initial process of the change, but the

reason and mechanism behind the specific formation were not discussed in this study. Furthermore, with the construction of increasingly water conservancy facilities in plain areas and the arrangement of sluices and pumps in the confluence of urban channels, the phenomena of drastic change in discharge ratios are progressively common due to convenient flow regulation [32,33]. Confluences are important nodes [29] that controls the connectivity and fluidity of river networks. A slight morphological change in these zones may trigger the "butterfly effect". By influencing the flow structure and topography at the junctions, and then affecting the material migration of the whole river network, the habits of organisms are therefore changed, thus affecting the ecosystem of the entire watershed. Therefore, studying the confluence hydraulics under drastically varying discharge ratios has special significance for plain river networks and urban channels management.

Recently, with the continuous improvement of experimental equipment, such as Acoustic Doppler Current Profiler (ADCP), increasingly related researches have been reported in field measurement [31,34,35]. Compared with field measurements, laboratory experiments have more flexible and changeable case settings, and are generally less costly and less labor-intensive. In laboratory tests, the control variable method is often used to make it clearer to set different working conditions for the required experimental purposes [12–15,22–24]. The application of ADV has significantly improved the level of velocity measurement in the laboratory channel experiments, especially in the measurement of turbulence [36–38]. These advantages of laboratory tests are the main reasons why it is used as a research method in this paper. In our study, the flow structure and topography were analyzed through three-dimensional velocity field of section and three-dimensional interpolation of terrain. The confluence area was the key research area, so the section layout was more intensive. However, there was still a lack of more detailed data to draw the contour map of 3D velocity field. Generally, the movable bed flumes determined that the bed surface reached equilibrium by the equal rate of sediment inflow and outflow [14,22–24]. Nonetheless, our channel device was a fixed bed structure, so there was no sediment input. The assessment of bed surface balance was mainly determined by the output of sediments downstream of the main tank. The equilibrium was reached when the sediment output at the downstream water outlet was zero. In the specific experiment process, the equilibrium state can be guaranteed by the 20–24 h scouring [39,40].

Our research was mainly to simulate the confluence hydraulics of drastically varying discharge ratios when the flood comes, so the transition time between the flow patterns was very short, which was in line with the actual situation [41]. However, in natural rivers, especially large-scale rivers with large width-depth ratio, the sharp change in the discharge ratio may not occur even if the flood comes and the transition time between flow patterns is relatively longer [42]. A longer transition time should be set for the laboratory simulation of varying discharge ratios of large natural river confluences. Therefore, our study is more suitable for small- and medium-sized rivers and urban channels with relatively small width-depth ratio. Additionally, most of the existing laboratory flumes are confluences formed by straight channels, while natural rivers usually bend [43]. The bend setting of flumes device to better simulate the confluence of curves of real rivers is worth improving.

Interestingly, the combination of confluence hydraulics with environmentology and ecology is gradually coming into the field of vision of related researchers. Many studies reported the transport of pollutants [30] and the assessment of biodiversity in channel confluences [44], which provides some reference and help for the river ecosystem protection. Our research has not yet involved the combination with environmental and ecological factors. Whether the drastically varying discharge ratios will trigger the butterfly effect, that is, the impact on the ecosystem of the entire river network by influencing the situation of the junctions needs to be further studied.

## 5. Conclusions

This article reported the 3D flow structure of three cases and the bed topography of two cases under the change in the discharge ratio in a 90-degree channel confluence.

Several important conclusions are drawn as follows:

(1) The drastic change in discharge ratio causes secondary scouring to the equilibrium bed topography in the confluence area. The bed surface at the sand hole and sand bar drops and the sediment is transported downstream. In this experiment, although local sand hole and sand ridge were formed in the flow recovery zone downstream, the results may be more suitable for urban channel confluence with relatively large width-depth ratio and small- and medium-sized natural channel confluences.

(2) The change in the bed topography has the greatest effect on the U-field, and the area of the maximum velocity zone is enlarged. Although the confluence at the same discharge and distinct topography has a smaller maximum velocity, it exhibits greater vortex and turbulence, which is the main cause of secondary erosion.

(3) For the V-, W-, and k-fields, the flow field is mainly controlled by the discharge and the corresponding ratio, whereas the change in topography only has a secondary effect in local areas. In addition, the distortion of the shear layer is caused by the change in the discharge ratio and has nothing to do with the topography.

**Author Contributions:** Conceptualization, investigation, validation, data curation, writing—original draft preparation, writing—review and editing, visualization, Z.Z.; methodology, formal analysis, resources, supervision, project administration, Y.L. All authors have read and agreed to the published version of the manuscript.

**Funding:** This research received no external funding.

**Data Availability Statement:** The data presented in this study are available on request from the corresponding author. The data are not publicly available due to privacy or ethical restrictions.

**Conflicts of Interest:** The authors declare no conflict of interest.

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
