# Peer review of "An Experimental Study on the Influence of Drastically Varying Discharge Ratios on Bed Topography and Flow Structure at Urban Channel Confluences"

_water, doi:10.3390/w13091147_

Round 1
Reviewer 1 Report
Please, see the attached document.

Author Response
Thank you for your comments concerning our manuscript entitled “Butterfly effect caused by small changes at at river confluences”. (ID: water-1138406). Those comments are all valuable and very helpful for revising and improving our paper, as well as the important guiding significance to our researches. We have studied comments carefully and have made correction which we hope meet with approval. Please see the attachment for details.

Reviewer 2 Report
The authours research focus is to investigate flow structure caused by bed topography variation under a rapid change in flow condition. The manuscript Introduction and Methodology sections need to be restructured.
The first appears not clear to reader and references not sufficiently support the overall experimental study.
The second does not provide adequate information regarding measurement equipment and experimental model setup (e.g. axis coordinate missing in transects location figure).
Results plot must be improved with contour maps which would assist reader to visualize study findings.
Discussion section may be benefit from these appropriate edits.
References list is limited.
I suggest Editors to consider major revisions to the manuscript even tough has interesting insights.

Author Response

(The authors gave the same response as above.)

Round 2
Reviewer 1 Report
I appreciate the work done by the Authors. The manuscript has been improved also according to my previous comments.
I suggest including the following references in the Introduction, because I found it still poor with respect to the extensive existing literature on this topic:
- Leite Ribeiro, M.; Blanckaert, K.; Roy, A.; Schleiss, A.J. Flow and sediment dynamics in channel confluence. J. Geophys. Res. Earth Surf. 2012, 117, doi:10.1029/2011JF002171.
- Rhoads, B.L.; Sukhodolov, A.N. Field investigation of three-dimensional flow structure at stream confluence: 1. Thermal mixing and time-averaged velocities. Water Resour. Res. 2001, 37, 2393–2410.
- Constantinescu, G.; Miyawaki, S.; Rhoads, B.; Sukhodolov, A.; Kirkil, G. Structure of turbulent flow at a river confluence with momentum and velocity ratios close to 1: Insight provided by an eddy-resolving numerical simulation. Water Resour. Res. 2011, 47, doi:10.1029/2010WR010018.
- Schindfessel, L.; Creëlle, S.; De Mulder, T. How Different Cross-Sectional Shapes Influence the Separation Zone of an Open-Channel Confluence. J. Hydraul. Eng. 2017, 143, 04017036.
- Penna, N., De Marchis, M., Canelas, O. B., Napoli, E., Cardoso, A. H., & Gaudio, R. (2018). Effect of the junction angle on turbulent flow at a hydraulic confluence. Water, 10(4), 469.
- Shakibainia, A.; Tabatabai, M.R.M.; Zarrati, A.R. Three-dimensional numerical study of flow structure in channel confluences. Can. J. Civ. Eng. 2010, 37, 772–781.
Regarding the methodology used to process the data (both velocity and topography data), it is required to explain in details the procedures. For example, it is not sufficient saying that the velocity data and topography data were post-processed by Tecplot 360 software.
The Authors said that they did not replace the spikes of the velocity signal, but this information must be included in the manuscript.
Finally, also the calculation of the turbulent kinetic energy must be discussed.
Author Response

(The authors gave the same response as above.)

Reviewer 2 Report
Please see attached PDF file for minor rivisions.

Author Response

(The authors gave the same response as above.)
